# Improving Chinese Pop Song and Hokkien Gezi Opera Singing Voice Synthesis by Enhancing Local Modeling

**Peng Bai[1,2,*], Yue Zhou[1,2,*], Meizhen Zheng[1,2], Wujin Sun[2,3] and Xiaodong Shi[1,2,3,†]**

[1]Department of Artificial Intelligence, School of Informatics, Xiamen University, China
[2]Key Laboratory of Digital Protection and Intelligent Processing of Intangible Cultural Heritage of Fujian and Taiwan (Xiamen University), Ministry of Culture and Tourism, China
[3]Institute of Artificial Intelligence, Xiamen University, China
{baipeng,zhouyue1,midon,sunwujin}@stu.xmu.edu.cn   mandel@xmu.edu.cn

## Abstract

Singing Voice Synthesis (SVS) strives to synthesize pleasant vocals based on music scores and lyrics. The current acoustic models based on Transformer usually process the entire sequence globally and use a simple L1 loss. However, this approach overlooks the significance of local modeling within the sequence and the local optimization of the hard-to-synthesize parts in the predicted mel-spectrogram. Consequently, the synthesized audio exhibits local incongruities (*e.g.*, local pronunciation jitter or noise). To address this problem, we propose two methods to enhance local modeling in the acoustic model. First, we devise a nearest neighbor local attention, where each phoneme token focuses only on the adjacent phoneme tokens located before and after it. Second, we propose a phoneme-level local adaptive weights loss function that enables the model to focus more on the hard-to-synthesize parts of the mel-spectrogram. We verify the universality of our methods on public Chinese pop song and Hokkien Gezi Opera datasets. Extensive experiments demonstrate the effectiveness of our methods, resulting in significant improvements in both objective and subjective evaluations when compared to the strong baselines. Our code and demonstration samples are available at https://github.com/baipeng1/SVSELM.

## 1 Introduction

Singing Voice Synthesis (SVS) converts the lyrics into natural and humanlike voice audio according to the music scores (Yi et al., 2019)[1]. Due to its promising application in fields such as virtual singer and music education, SVS has attracted the attention of a large number of researchers recently (Hono et al., 2019; Lu et al., 2020; Gu et al., 2021; Liu et al., 2022; He et al., 2023). SVS systems generally consist of an acoustic model and

a vocoder. The acoustic model converts music scores and lyrics into acoustic features (*e.g.*, mel-spectrogram), and the vocoder synthesizes audio waveform from acoustic features (Liu et al., 2022).

Recently, Transformer (Vaswani et al., 2017) has been widely used in sequence modeling tasks. The acoustic models based on Transformer have showed great performance, including FFT-NPSS (Blaauw and Bonada, 2020), XiaoiceSing (Lu et al., 2020), DeepSinger (Ren et al., 2020), FFT-Singer (Liu et al., 2022). However, these models still exhibits local incongruity in the synthesized audio, which is characterized by local pronunciation jitter or noise. Local incongruity will bring negative experiences to listeners, so this is a problem that urgently needs to be solved. Lee et al. (2021) also focus on local incongruity problem. In order to improve the accuracy of the local pronunciation, they added a postnet to the model and used adversarial training methods, where the voicing-aware discriminator was used to capture the harmonic features of vocal segments and the noise components of silent segments. Unlike their approaches, we abstain from employing post processing networks or adversarial training methods. Instead, we address this problem from the perspective of enhancing local attention and refining loss function.

Some studies (Yang et al., 2020; Watzel et al., 2021; Zhu et al., 2021; Cao et al., 2021) discovered that incorporating additional local attention can enhance model performance in text to speech (TTS) and automatic speech recognition (ASR) tasks. In addition, some studies (Lin et al., 2017; George and Marcel, 2021; Li et al., 2022) used the local focus loss function in the image reconstruction task to improve the reconstruction effect. Inspired by the above works, in order to address the problem of local incongruity in acoustic models based on Transformer, we propose two methods to enhance local modeling. First, the attention mechanism in these acoustic models is the global contextual

---

[*]Equal contribution.
[†]Corresponding author.
[1]The music score referred to in this paper does not include lyrics.

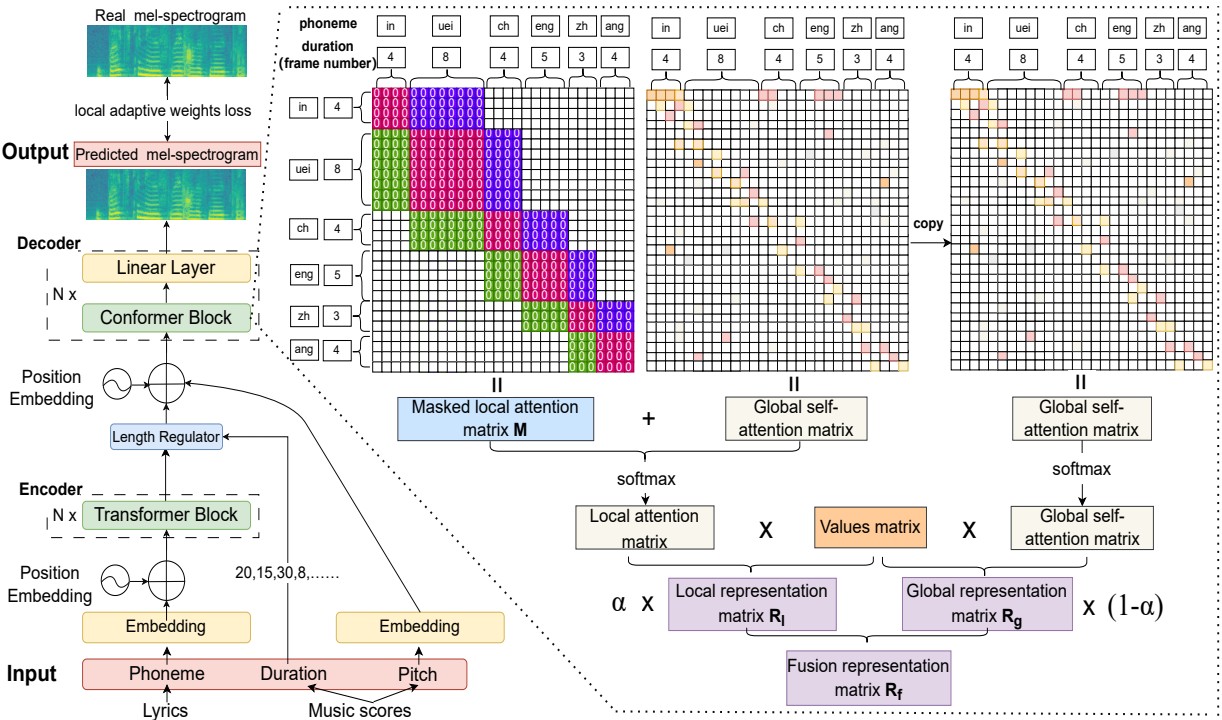

Figure 1: The structure of the our model.

self-attention of the entire sequence, and such an attention mechanism may disperse the local attention in the sequence. So we devise a nearest neighbor local attention to only focus on the phoneme tokens that are close to each other in a short distance. Second, the loss function in these acoustic models is generally a simple L1 loss. It optimizes each part of the mel-spectrogram with equal weights. This will lead to the hard-to-synthesize parts of the mel-spectrogram still being in a difficult position. So we propose a phoneme-level local adaptive weights loss to mainly optimize the hard-to-synthesize parts.

The main contributions of this paper can be summarized as follows:

- We devise a nearest neighbor local attention to only focus on the adjacent phoneme tokens located before and after the target phoneme token in a short distance.

- We propose a novel phoneme-level local adaptive weights loss to optimize the local hard-to-synthesize parts in the predicted mel-spectrogram.

- The extensive experiments on public Chinese pop song and Hokkien Gezi Opera datasets have demonstrated the effectiveness and uni-

versality of our local modeling enhancement methods.

## 2 Methods

In Section 2.1, we first introduce an overview of our model. We then introduce the nearest neighbor local attention method in Section 2.2, and finally introduce the local adaptive weights loss method in Section 2.3.

### 2.1 Overview of Model

As shown in Figure 1, our model consists of an encoder, a length regulator, and a decoder.

**Encoder** The encoder in our model is the same as the Transformer block in the FastSpeech2 (Ren et al., 2021). The input of the encoder is the lyrics phoneme. After passing through the embedding layer, the phoneme sequence is inputted to the encoder with position embedding.

**Length Regulator** The length regulator expands the phoneme-level sequence into the frame-level sequence. The duration of phonemes has been obtained during the data processing stage.

**Decoder** The decoder in our model is Conformer (Gulati et al., 2020) block with linear layer. Conformer is a convolution-augmented Transformer. The input of the decoder is the output representation of the encoder, pitch embedding, du-

ration information, and position embedding. Pitch and duration are the important content of the music score. After the entire representation is processed by the decoder, the output of the decoder is a predicted mel-spectrogram.

## 2.2 Nearest Neighbor Local Attention

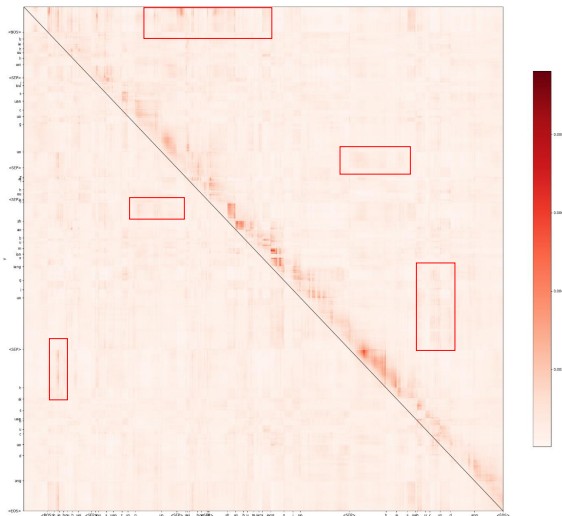

Figure 2: The visualization result of the global self-attention of the first Transformer block in the FFT-Singer decoder.

**Local Attention Layer** In the SVS task, each word token is composed of one or more phoneme tokens. Each phoneme token consists of multiple frame tokens. As shown in Figure 2, from the visualization result of the global self-attention of the first Transformer block in the FFT-Singer decoder, it can be seen that the existing global self-attention primarily focuses on the phoneme tokens area adjacent to the current phoneme token. The overall attention still exhibits a predominantly diagonal distribution. In addition, we also see that some phoneme tokens tend to focus on distant areas, and we mark them with red boxes. Based on the above observations, we think that the local attention of each phoneme token is insufficient. So we add a nearest neighbor local attention layer in the decoder to enhance the local attention of each phoneme. We first construct a nearest neighbor local attention matrix to only focus on the phoneme tokens that are close to each phoneme token in a short distance. We then use a gated unit mechanism to fuse local attention representation with global self-attention representation.

Considering that singers usually focus on the word they are currently singing in performance in-

stead of paying too much attention to other words in the entire lyrics at the same time, so we specially devise a nearest neighbor local attention that only focuses on the previous and next phoneme token. We first need to construct a masked phoneme-level local attention matrix. Figure 1 demonstrates the constructed masked local attention matrix, in which the frame tokens area corresponding to the current phoneme token is shown in pink, and the frame tokens area corresponding to the previous phoneme token is green, and the frame tokens area corresponding to the next phoneme token is purple.

Specifically, $P = \{p_1, \cdots, p_i, \cdots, p_n\}$ is a phoneme sequence, where $n$ is the number of phoneme in a certain sample. We define the current phoneme token ID as $i$, the locally focused phoneme token number before the current phoneme token as $l$, and the locally focused phoneme token number after the current phoneme token as $r$. In the masked phoneme-level local attention matrix $M$, the attended phoneme tokens are set to 0, while the rest phoneme tokens are negative infinity. Thus $M$ can be represented as follows:

$$M_{p,g} = \begin{cases} 0, & p, g \in [i-l, i+r] \\ -\infty, & \text{otherwise} \end{cases}, \quad (1)$$

where both $p$ and $g$ are the phoneme ID of the entire phoneme-level representation sequence.

We add masked local attention matrix $M$ to the global self-attention matrix. In fact, $M$ acts as a mask role, which preserves the content of the global self-attention matrix corresponding to the position with the content 0 in the matrix $M$. Therefore, the formula for sequence local representation $R_l$ is as follows:

$$R_l = \text{softmax}(M + \frac{QK^\top}{\sqrt{d_k}})V, \quad (2)$$

where $Q$, $K$ and $V$ are the query, key and value for sequence. $d_k$ is the dimension of keys. $\text{softmax}(\cdot)$ is a normalization function.

**Fusion of local and global representation** As shown in Figure 1, on the basis of local representation $R_l$, we use a gated unit coefficient $\alpha$ to fuse it with the original global representation $R_g$. In this way, we will obtain a fusion representation $R_f$ of local and global representation. The representation of each phoneme token in the $R_f$ is strengthened.

The formula for $R_g$ is as follows:

$$R_g = \text{softmax}(\frac{QK^\top}{\sqrt{d_k}})V, \quad (3)$$

where $Q$, $K$, $V$ and $d_k$ are similar to Eq. 2.

The formula for $R_f$ is as follows:

$$R_f = \alpha R_l + (1 - \alpha)R_g, \qquad (4)$$

where $\alpha$ is a learnable coefficient, and $\alpha \in [0, 1]$.

The formula for $\alpha$ is as follows:

$$\alpha = \mathrm{sigmoid}(W([R_l; R_g])), \qquad (5)$$

where $\mathrm{sigmoid}(\cdot)$ is an activation function. $W$ is a fully connected layer. $[;]$ is a concatenation operation and concats the channel dimension of $R_l$ and $R_g$.

## 2.3 Local adaptive weights loss

Ordinary L1 loss optimizes each part of the mel-spectrogram with equal weights, so it will result in hard-to-synthesize parts in the predicted mel-spectrogram still difficult to synthesize. We are driven by the motivation to optimize each phoneme region, with particular emphasis on the hard-to-synthesize parts within the predicted mel-spectrogram. As shown in Figure 3, we propose a phoneme-level local adaptive weights loss to replace L1 loss. Specifically, We calculate the phoneme-level adaptive confidence based on the phoneme region, and the confidence scores represent the synthesis quality of each phoneme region in the current mel-spectrogram. We also normalize the phoneme-level adaptive confidence scores to phoneme-level adaptive weights, which can dynamically update the weight of the phoneme region in the predicted mel-spectrogram. We finally multiply the adaptive weights by the values of different phoneme regions in the mel-spectrogram.

**Phoneme-level Adaptive Confidence** We use the confidence of phoneme-level mel-spectrogram to determine the emphasis of the model's learning, which quantifies whether the predicted mel-spectrogram phoneme region is close to or far from the real mel-spectrogram. The confidence scores $m_k$ are calculated as follows:

$$m_k = \mathrm{Ave}\left(\left| M_p'(i, j) - M_p(i, j) \right|\right), \qquad (6)$$

where $k$ is the phoneme ID and its range is $1, 2, \cdots, n$. $M_p'(i, j)$ is the phoneme-level predicted mel-spectrogram. $M_p(i, j)$ is the phoneme-level real mel-spectrogram. $i$ is the frame ID, and $j$ is the mel bins ID. $\mathrm{Ave}(\cdot)$ is an operation that averages the mel-spectrogram of the phoneme region.

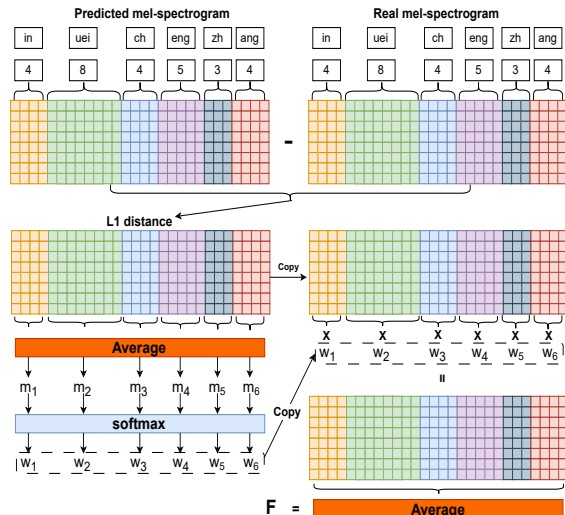

Figure 3: The schematic diagram of the calculation process for local adaptive weights loss.

**Phoneme-level Adaptive Weight** We use the $\mathrm{softmax}$ function to normalize the confidence scores and obtain the adaptive weights $\omega_k$:

$$\omega_k = \frac{e^{m_k}}{\sum_{z=1}^{n} e^{m_z}}, \quad k = 1, 2, \cdots, n. \qquad (7)$$

Finally, we multiply the phoneme-level adaptive weights by the original L1 loss matrix according to the phoneme region. The formula for the local adaptive weights loss value $F$ is as follows:

$$F = \frac{1}{MN} \sum_{i=0}^{M-1} \sum_{j=0}^{N-1} \omega_k \left| M_p'(i, j) - M_p(i, j) \right|, \qquad (8)$$

where $M$ is the frame number of the sample, and $N$ is the number of mel bins.

## 3 Experiments

### 3.1 Datasets

**PopCS Dataset** PopCS (Liu et al., 2022) is a Chinese Mandarin pop singing dataset recorded by a female professional singer in a recording studio. This dataset has a total of approximately 5.89 hours and contains 117 Chinese songs, with each audio is sampled at 24kHz and quantized using 16 bits.

**Gezi Opera Dataset** Gezi Opera is one of the traditional local operas of the Hokkien dialect in China, and is a national intangible cultural heritage. This dataset is recorded by 5 professional Gezi Opera singers using mobile phones, consisting of 3 actresses and 2 male actors. The total duration is approximately 4.5 hours. Each audio is sampled at

48kHz and quantized using 16 bits. This dataset is built by our team [2].

## 3.2 Comparative models

- The **Baseline** model is the FFT-Singer (Liu et al., 2022) based on FastSpeech2. The encoder and decoder are Transformer blocks. The training loss is L1 loss.
- The **Baseline-T+C** is the model that replaces the Baseline decoder with Conformer blocks. The training loss is L1 loss.
- The **Baseline-T+C+A** model on the basis of **Baseline-T+C** replaces the training loss with our local adaptive weights loss.
- The **Baseline-T+C+L** model adds our nearest neighbor local attention to the the **Baseline-T+C** model decoder. The training loss is L1 loss.
- The **Baseline-T+C+A+L** model is our final model. It adds our nearest neighbor local attention to the **Baseline-T+C+A** model decoder. The training loss is local adaptive weights loss.
- The **N-Singer** (Lee et al., 2021) model is a Korean SVS model that focuses on addressing the accuracy of pronunciation in local incongruity problems. It includes a Transformer-based mel-generator, a convolutional network-based postnet, and voicing-aware discriminators [3].
- The **Baseline+GAN** model is an adversarial training method that we add to the Baseline, and the method used to generative adversarial network (GAN) comes from HiFiSinger (Chen et al., 2020).

## 3.3 Model configuration

Our model is modified based on FastSpeech2. The encoder is Transformer block and the decoder is Conformer block. In terms of the global configuration of the model, the audio sampling rate is 24kHz. Because we extract the real mel-spectrogram from real audio, the length of the Fast Fourier Transform window is 512 and the hop length is 128. The number of mel bins is 80. The representation dimension of the input token and channels are all 256. Meanwhile, the dimension of attention is also 256. The encoder has the same settings as the en-

coder in FastSpeech2. Both the Transformer block and Conformer block have multi-head attention. The number of head in multi-head attention is 4. The number of block is set to 4 in encoder and decoder. The Transformer block of the encoder contains 1d convolution, and the size of the kernel is 9. The Conformer block of the decoder contains a depthwise 1d convolutional layer with a kernel size of 31. In addition, there are two pointwise convolutional layers with kernel size of 1. The Conformer block includes gated linear unit activation and swish (Ramachandran et al., 2018) activation. Our SVS model is a two-stage model. Our proposed methods are added to an acoustic model, so we also choose a HiFi-GAN (Kong et al., 2020) singing model pre-trained by the DiffSinger open-source project as the vocoder to synthesize audio. We use the AdamW (Loshchilov and Hutter, 2017) optimizer. Our maximum training steps are 160k. We train the model on a single A40 GPU.

## 3.4 Evaluation metrics

In order to evaluate the performance of our proposed method, we conduct objective and subjective evaluations. In objective evaluation, we use Mean Cepstral Distortion (MCD) and Mean Spectral Distortion (MSD) to evaluate the timbral distortion of synthesized audio. We also select Gross Pitch Error (GPE), Voicing Decision Error (VDE), and $F_0$ Frame Error (FFE) (Chu and Alwan, 2009) to evaluate the $F_0$ track in synthesized audio. We use the code implemented in fairseq [4] for objective evaluation.

For subjective evaluation, we conduct a Mean Opinion Score (MOS) for both real and synthesized audio in the test set. MOS mainly evaluates the human subjective perception of audio naturalness. The rating range of MOS is from 1 to 5. 1 represents the lowest naturalness, and 5 represents the highest naturalness. 10 volunteers participated in the evaluation without disclosing the audio source.

# 4 Results

This section shows the results of the experiments. Section 4.1 is the main comparison and analysis of the overall objective and subjective results. Section 4.2 is the experiments about the best number selection for the previous and next masked phoneme tokens. Section 4.3 explores how our

---

[2]This dataset can be authorized by contacting us and used only for scientific research.

[3]Due to the lack of open official code for N-Singer, we reproduced it.

[4]https://github.com/facebookresearch/fairseq/blob/main/examples/speech_synthesis/docs/ljspeech_example.md

| Dataset | Model | MCD(dB)↓ | MSD(dB)↓ | GPE(%)↓ | VDE(%)↓ | FFE(%)↓ |
|---|---|---|---|---|---|---|
| PopCS | Baseline | 3.4065 | 1.7164 | 0.74 | 3.63 | 4.05 |
| | Baseline-T+C | 3.2646 | 1.638 | 0.75 | 3.75 | 4.18 |
| | Baseline-T+C+A | 3.062 | 1.5475 | 0.75 | 3.46 | 3.89 |
| | Baseline-T+C+L | 2.9991 | 1.5452 | **0.64** | 3.83 | 4.2 |
| | Baseline-T+C+A+L | **2.8735** | **1.4809** | 0.65 | **3.3** | **3.67** |
| | N-Singer | 2.9561 | 1.5523 | 3.29 | 3.85 | 4.95 |
| | Baseline+GAN | 3.0324 | 1.645 | 5.88 | 4.05 | 8.81 |
| Gezi Opera | Baseline | 3.4694 | 1.7498 | 1.5 | 3.7 | 4.81 |
| | Baseline-T+C | 3.3911 | 1.6924 | 1.75 | 3.67 | 4.97 |
| | Baseline-T+C+A | 3.0314 | 1.5459 | 1.75 | 3.78 | 5.07 |
| | Baseline-T+C+L | 3.017 | 1.5595 | 1.47 | 3.59 | 4.71 |
| | Baseline-T+C+A+L | **2.931** | **1.5144** | **1.34** | **3.57** | **4.57** |
| | N-Singer | 3.021 | 1.582 | 5.88 | 3.69 | 7.82 |
| | Baseline+GAN | 3.0728 | 1.937 | 6.94 | 4.12 | 9.86 |

Table 1: The objective evaluation results on the PopCS dataset and Gezi Opera dataset. The **Baseline** model is the **FFT-Singer**. **-T** means removing Transformer blocks from the decoder of the Baseline model. **+C** means adding Conformer blocks to the decoder of the Baseline model. **+A** means that the loss function is replaced by the default L1 loss with the local adaptive weights loss. **+L** means adding the nearest neighbor local attention in the decoder.

methods can flexibly combine with other models, such as DiffSinger. Section 4.4 is a case study, which visually demonstrates the effectiveness and universality of our proposed methods.

### 4.1 Main result analysis

Table 1 shows the results of the objective evaluation metrics of the models on the PopCS dataset and the Gezi Opera dataset. For every dataset, the first line is the result of the Baseline model. In the second line, when we replace the Baseline decoder with a Conformer block, the results will decrease. This result validates that the performance of Conformer blocks surpasses the Transformer blocks in the SVS task. The convolution module in the Conformer block is more effective. In the third line, we can see that after replacing the ordinary L1 loss with the local adaptive weights loss, the metrics continue to decrease, which also confirms the effectiveness of our proposed loss. The reason is that the local adaptive weights loss can dynamically expand the weights of hard-to-synthesize parts in the mel-spectrogram, making the model to focus on optimizing the hard-to-synthesize parts. In the fourth and fifth lines, we add the nearest neighbor local attention to the previous model, and we can see a further decrease in metrics. Especially in the fifth line, after adding both the nearest neighbor local attention and local adaptive weights loss, the metrics achieve the lowest value. These two lines of results indicate that the nearest neighbor local attention

method has worked. The N-Singer in the sixth line and the Baseline+GAN in the seventh line are all GAN-based methods. We can see that the objective evaluation metrics of these two models, especially GPE and FFE, are not ideal. However, we find that the increase of GPE and FFE did not cause a significant decrease in subjective perception. It is necessary to conduct subjective evaluation and we cannot rely too much on objective evaluation. We should combine these two aspects for comprehensive evaluation.

| Dataset | Model | MOS↑ |
|---|---|---|
| PopCS | Ground Truth | 4.43±0.08 |
| | Baseline | 3.55±0.12 |
| | Baseline-T+C+A+L | **3.71**±0.11 |
| | N-Singer | 3.65±0.1 |
| | Baseline+GAN | 3.63±0.13 |
| Gezi Opera | Ground Truth | 4.33±0.09 |
| | Baseline | 3.46±0.15 |
| | Baseline-T+C+A+L | **3.61**±0.12 |
| | N-Singer | 3.58±0.11 |
| | Baseline+GAN | 3.51±0.12 |

Table 2: The MOS results on the PopCS and the Gezi Opera dataset. MOS is reported with 95% confident intervals.

Table 2 shows the results of the subjective evaluation metrics of the models on the PopCS dataset and the Gezi Opera dataset. As shown in the PopCS dataset, our final model achieves the highest MOS

value of 3.71. As shown in the Gezi Opera dataset, our final model achieves the highest MOS value of 3.61. In the analysis of the synthesized samples, we also find that the two GAN-based methods, N-Singer and Baseline+GAN, have positive effects in terms of noise, but the GAN-based methods sometimes suffer from pitch inaccuracies on the Gezi Opera dataset, which is proved by the GPE and FFE metrics in Table 1.

On the two datasets of SVS tasks, the Baseline model adding our methods achieves the best results in both objective and subjective evaluations, which fully demonstrates the effectiveness and universality of the two local modeling enhancement methods we proposed.

## 4.2   Number of phoneme tokens selection

This section is how to determine the best number of phoneme tokens before and after the current phoneme token in the masked phoneme-level local attention matrix. We conduct experiments on the Baseline-T+C+A+L model in the PopCS dataset, employing seven different scenarios for number selection. As we defined in section 2.2, $l$ is the number of locally focused phoneme tokens before the current phoneme token, and $r$ is the number of locally focused phoneme tokens after the current phoneme token. The seven scenarios we set are "l=0 and r=0", "l=0 and r=1", "l=1 and r=0", "l=1 and r=1", "l=1 and r=2", "l=2 and r=1" and "l=2 and r=2". The reason for our setting is that we observe the self-attention matrix from Figure 2, it can be seen that roughly one to two phoneme tokens range before and after each current phoneme token are mainly being focused on. Finally, we select the optimal parameters through objective and subjective evaluation metrics.

| l and r | MCD(dB)↓ | FFE(%)↓ | MOS↑ |
|---------|----------|---------|------|
| l=0 r=0 | 2.93 | 3.77 | - |
| l=0 r=1 | 2.9026 | 3.87 | - |
| l=1 r=0 | 2.8581 | 3.86 | 3.65±0.13 |
| l=1 r=1 | 2.8735 | **3.67** | **3.71±0.11** |
| l=1 r=2 | **2.838** | 4.02 | 3.61±0.12 |
| l=2 r=1 | 2.8615 | 3.83 | - |
| l=2 r=2 | 2.883 | 3.89 | - |

Table 3: The MCD, FFE, and MOS results of Baseline-T+C+A+L model on the PopCS dataset. MOS is reported with 95% confident intervals.

As shown in Table 3, we can see that when "l=1 and r=2", MCD achieves the lowest at 2.838, but

FFE is the highest at 4.02. When "l=1 and r=1", MCD is 2.8735 and FFE is the lowest at 3.67. In subjective evaluation, "l=1 and r=1" is the highest at 3.71. Considering both objective and subjective evaluation results, we believe that under the premise of approximate MCD, we should focus on FFE and MOS. So we choose the setting of "l=1 and r=1". In this setting, it is possible to ensure that the attention is focused on the initial and final phonemes of each word. Our main experiments adopt this setting.

## 4.3   Method flexibility

The two methods we proposed can be flexibly combined with other models. As long as the original acoustic model utilizes global self-attention and L1 loss, our methods can be flexibly applied to these models to improve performance. We validate the performance when using the DiffSinger model (Liu et al., 2022). DiffSinger is a SVS model based on the diffusion model. We have set two scenarios, one diffusion condition is the mel-spectrogram predicted by FFT-Singer and the other diffusion condition is the mel-spectrogram predicted by FFT-Singer adding our methods.

| Dataset | Model | MOS↑ |
|---------|-------|------|
| PopCS | Ground Truth | 4.43±0.08 |
| | DiffSinger | 3.86±0.12 |
| | DiffSinger+Our | **3.91±0.11** |
| Gezi Opera | Ground Truth | 4.33±0.09 |
| | DiffSinger | 3.82±0.1 |
| | DiffSinger+Our | **3.86±0.12** |

Table 4: The MOS results on PopCS and Gezi Opera datasets. MOS is reported with 95% confident intervals. **Diffsinger** represents diffusion based on the mel-spectrogram predicted by FFT-Singer. **DiffSinger+Our** represents the diffusion of DiffSinger based on the mel-spectrogram predicted by FFT-Singer adding our methods.

As shown in Table 4, we can see that after adding our methods, the MOS score of DiffSinger+Our is higher than that of the basic DiffSinger. This demonstrates that the audio synthesized by DiffSinger on the basis of FFT-Singer adding our methods is better. This further validates the flexibility and practicality of our methods. We also find that the DiffSinger+Our model can indeed solve some of the local incongruity problems that exist in the DiffSinger model, especially local pronunciation jitter.

## 4.4 Case study

In order to more intuitively demonstrate the local modeling enhancement effects of our proposed methods in the mel-spectrogram prediction process, Figure 4 and Figure 5 respectively show the mel-spectrogram visualization results of a certain sample on the PopCS dataset and the Gezi Opera dataset by different models.

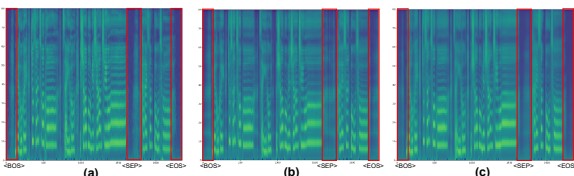

Figure 4: The mel-spectrogram visualization of the same sample in the PopCS dataset. **(a)** is the real mel-spectrogram of the sample. **(b)** is the mel-spectrogram predicted by the Baseline model. **(c)** is the mel-spectrogram predicted by the Baseline-T+C+A+L model.



Figure 5: The mel-spectrogram visualization of the same sample in the Gezi Opera dataset. **(a)** is the real mel-spectrogram of the sample. **(b)** is the mel-spectrogram predicted by the Baseline model. **(c)** is the mel-spectrogram predicted by the Baseline-T+C+A+L model.

As shown in Figure 4, the phoneme sequence of the lyrics is "<BOS> b ie | h ou | h uei <SEP> j iou | s uan | c uo | g uo <SEP> z ai | i | h ou <SEP> n i | sh ao | b u | m ian | x iang | q i | uo <SEP> h ai | s uan | b u | c uo <EOS>", and there are breathing or silent segments in "<BOS>", "<SEP>", and "<EOS>" token segments. We highlight these parts with red boxes in the mel-spectrogram. In Figure 4(b), these three phoneme segments are relatively noisy compared to Figure 4(a). In Figure 4(c), we can see that the mel-spectrogram predicted by our methods is closer to the real mel-spectrogram and has less noise at these three segments.

In Gezi Opera, in order to express a certain emotion, artistic modifications are often made to the pronunciation of the final or after the final. Artistic modifications usually involve long-term multi rhythmic singing. As shown in Figure 5, the

phoneme sequence of the lyrics is "sp ua sp $k^h$ o sp m ia e h i ien an an $ts^h$ e sp e e e sp e e e e e e e e e sp". At the following segment "e e e e e e e e e", the medium and high frequency bands of the corresponding parts in the mel-spectrogram are marked with red boxes. We can see that the predicted content in Figure 5(c) is more detailed than that in Figure 5(b). The Figure 5(c) is closer to the real mel-spectrogram in Figure 5(a). This result also reflects that our methods can indeed improve the local quality of the predicted mel-spectrogram.

## 5 Related Works

### 5.1 Singing voice synthesis

At the end of the 1950s, the earliest computer music project studied by Bell Labs realized the SVS, and a representative physical acoustic model is Kelly and Lochbaum (Cook, 1996). After the development of traditional methods such as unit splicing and statistical parameters, the current mainstream method is based on deep learning.

With the rapid development of deep learning (Lin et al., 2023), the implementation of SVS research mainly adopts various neural network architectures. The current SVS research can be divided into integrated end-to-end and fully end-to-end model. The integrated end-to-end SVS system consists of an acoustic model and a vocoder. XiaoiceSing (Lu et al., 2020), DeepSinger (Ren et al., 2020), ByteSing (Gu et al., 2021), HiFiSinger (Chen et al., 2020), XiaoiceSing2 (Wang et al., 2022) are all integrated end-to-end models. Among the above models, the acoustic model in ByteSing is based on a recurrent neural network, while the rest acoustic models are all based on Transformer. HiFiSinger and XiaoiceSing2 adopt adversarial training. The vocoders used in these models are usually WORLD(Morise et al., 2016), HiFi-GAN (Kong et al., 2020) or MelGAN (Kumar et al., 2019), etc. Liu et al. (2022) designed an acoustic model named DiffSinger based on the diffusion probabilistic model. They proposed a shallow diffusion mechanism to improve audio quality and accelerate inference. Visinger(Zhang et al., 2022a) and Visinger2 (Zhang et al., 2022b) are fully end-to-end models, and the acoustic model is trained together with the vocoder. This type of model can avoid the problem of error accumulation.

In our work, we propose two local modeling enhancement methods for the acoustic model based on Transformer.

## 5.2 Local modeling enhancement

In the research of TTS, Yang et al. (2020) proposed two local modeling enhancement methods to improve the performance of models based on the self-attention mechanism. One is the enhancement of local relative position perception representation for sequence representation. Another approach is learnable gaussian bias to enhance local representation in self-attention. In the research of ASR, some research works (Watzel et al., 2021; Zhu et al., 2021; Cao et al., 2021) also enhance recognition accuracy by strengthening local modeling. In natural language processing research, in order to enhance the local attention of sequence, Zhang et al. (2020) added a syntax-guided self-attention layer to improve the Transformer's performance in reading comprehension tasks. Li et al. (2021) proposed a syntax-aware local attention method to improve BERT. In the task of facial photo sketch synthesis, Yu et al. (2023) proposed an additional local self-attention for local correlation. Local attention can achieve better synthesis results by integrating with global self-attention.

In this work, we enhance the local modeling ability of the acoustic model from two perspectives: adding local attention and designing local adaptive weights loss.

## 6 Conclusion

In the Chinese pop song and Hokkien Gezi Opera singing voice synthesis tasks, we propose two local modeling enhancement methods in acoustic model based on Transformer to improve the quality of the predicted mel-spectrogram. One method is to enhance local attention for each phoneme token in the decoder and fuse local attention representation with the original global self-attention representation. Another method involves employing a novel phoneme-level local adaptive weights loss to optimize the hard-to-synthesize parts of the predicted mel-spectrogram. We conduct extensive experiments on the Chinese pop song and Hokkien Gezi Opera datasets, and both objective and subjective evaluation metrics show the effectiveness and universality of our methods in enhancing local modeling for mel-spectrogram prediction. Our two methods are simple and practical, and can be flexibly incorporated into acoustic models based on Transformer or Conformer. In summary, our methods can greatly alleviate the local inconsistency problem in SVS tasks and improve the quality of synthesized audio. We are moving towards a better solution to completely solve this problem.

## Limitations

We propose two local modeling enhancement methods in the SVS acoustic model. The methods can effectively alleviate the problem of local incongruity in synthesized audio. However, our work still has some limitations. (1) The proposed nearest neighbor local attention representation in this paper only verifies the effectiveness of the fusion with the global self-attention representation. (2) The nearest neighbor local attention brings additional computational requirements and increases the demand for GPU resources. (3) We find that our methods cannot completely solve the problem of local incongruity in SVS. Our methods significantly improve the effect of the silent or breathing segments, and can provide some relief for other segments. Our method still has room for improvement, and we think that we can further control the high, medium, and low frequency bands in the mel-spectrogram locally through the loss function.

## Ethics Statement

We use two SVS datasets in the experiments. one is PopCS dataset and the other one is Gezi Opera dataset built by us. The PopCS dataset has been authorized by the owner. The Gezi Opera dataset is collected, organized, and produced by ourselves. We have also deleted the singer's personal information to ensure their privacy. We strictly adopt a blind evaluation mechanism during subjective evaluation.

## Acknowledgements

This work is supported by the Project of Technological Innovation 2030 "New Generation Artificial Intelligence" (No.2020AAA0107904) and the Major Scientific Research Project of the State Language Commission in the 13th Five-Year Plan (No. WT135-38). We thank the reviewers for their valuable comments. Special thanks to the actors and actresses who participated in the audio recording of Gezi Opera.

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
