# OpenReview forum: "Improving Chinese Pop Song and Hokkien Gezi Opera Singing Voice Synthesis by Enhancing Local Modeling"
_EMNLP/2023/Conference — EMNLP 2023 Main_

### Official Review · Reviewer_knLR · 2023-08-06

**Soundness:** 3

**Excitement:**

3: Ambivalent: It has merits (e.g., it reports state-of-the-art results, the idea is nice), but there are key weaknesses (e.g., it describes incremental work), and it can significantly benefit from another round of revision. However, I won't object to accepting it if my co-reviewers champion it.

**Paper Topic And Main Contributions:**

The paper proposes two local modeling enhancement methods in acoustic model to improve the quality of the predicted mel-spectrogram. One method is to fuse the local attention representation with the original global self-attention representation. Another method is to replace the L1 loss with a local adaptive weights loss. The experimental results on the Chinese pop song and Hokkien Gezi Opera datasets show that the proposed method can improve the quality of synthesized audio.

**Reasons To Accept:**

(1)The paper proposes two innovations. One innovation is nearest neighbor local attention. Another innovation is local adaptive weights loss.
(2)The experimental results of the paper prove the effectiveness of the proposed method. The ablation experiments verify the effectiveness of the two innovations of the method.
(3)The proposed method can be flexibly integrated on other models.


**Reasons To Reject:**

(1)The paper has some innovations, but the innovations are a bit small. One is to integrate local attention into global attention, and the other is to replace equal weights on the loss with adaptive weights.
(2)The proposed method is integrated on the DiffSinger model in Section 4.3, but the results show that the improvement is not significant. From the MOS results in Table4, the improvement is weak. It is recommended to add objective evaluation to further prove the improvement effect of the algorithm.


**Reproducibility:**

5: Could easily reproduce the results.

**Reviewer Confidence:**

5: Positive that my evaluation is correct. I read the paper very carefully and I am very familiar with related work.

**Typos Grammar Style And Presentation Improvements:**

(1)Section 2.3, "Specifically, We calculate the..." should be "Specifically, we calculate the...".
(2)Section 6 Limitations, "Our methods significantly improves the..." should be "Our methods significantly improve the...".
(3)Section 6 Ethics Statement, "the experiments. one is PopCS dataset..." should be "the experiments. One is PopCS dataset...".

---

> ### Author Rebuttal · Authors · 2023-08-28
>
> **Q1**:Typos Grammar Style And Presentation Improvements.
>
> **A1**:Thank you for pointing out these obvious casing and singular/plural grammar errors, which were caused by our careless writing. After carefully reviewing the paper, we have corrected the grammar errors that exist in the paper.
>
> **Q2**:The paper has some innovations, but the innovations are a bit small. One is to integrate local attention into global attention, and the other is to replace equal weights on the loss with adaptive weights.
>
> **A2**:Your evaluation is very relevant. The improvement workload for the FFT-Singer model in this paper is a bit small, but the two improvement methods proposed in this paper are both simple and effective, and our methods have good robustness and generalization. As far as we know, we constructed the Gezi Opera dataset for the first time in singing voice  synthesis research, and the process of constructing the dataset was very difficult and time-consuming. In addition, we validated the effectiveness of the methods and the effectiveness of the Gezi Opera dataset. We have truly expanded the scope of research on singing voice  synthesis.
>
> **Q3**:The proposed method is integrated on the DiffSinger model in Section 4.3, but the results show that the improvement is not significant. From the MOS results in Table4, the improvement is weak. It is recommended to add objective evaluation to further prove the improvement effect of the algorithm.
>
> **A3**:We have added objective evaluation results as shown in the table below.
> #### **Objective evaluation results.**
>
> | Dataset | Model | MCD(dB)$\downarrow$ | MSD(dB)$\downarrow$ | GPE(\%)$\downarrow$ | VDE(\%)$\downarrow$ | FFE(\%)$\downarrow$ |
> | --- | --- | --- | --- | --- | --- | --- |
> | PopCS | DiffSinger | 3.1452 | 1.7365 | **0.72** | **3.67** | **4.08** |
> | PopCS | DiffSinger+Our | **3.0564** | **1.6706** | 0.9 | 3.74 | 4.24 |
> | Gezi Opera | DiffSinger | 3.9807 | 2.0012 | 1.7 | 3.84 | 5.1 |
> | Gezi Opera | DiffSinger+Our | **3.9145** | **1.9815** | **1.64** | **3.67** | **4.89** |
>
> From the table, we can see that adding our method to DiffSinger resulted in a decrease in both MCD and MSD in the PopCS dataset, while there was almost no significant change in GPE, VDE and FFE. In the Gezi Opera dataset, all metrics have shown a certain decrease. Combining subjective and objective evaluation metrics, the overall improvement effect for DiffSinger is not significant. The actual situation is indeed so, because the DiffSinger model itself has good performance, and our method was not originally designed to improve the performance of the DiffSinger model. Although this method has a small improvement on the DiffSinger model, we did significantly improve the performance of FFT-Singer, which is the starting point of this study. Our purpose in conducting this experiment is to verify that our method is flexible and effective even when combined with the DiffSinger model.

---

### Official Review · Reviewer_DiTD · 2023-08-09

**Soundness:** 3

**Excitement:**

3: Ambivalent: It has merits (e.g., it reports state-of-the-art results, the idea is nice), but there are key weaknesses (e.g., it describes incremental work), and it can significantly benefit from another round of revision. However, I won't object to accepting it if my co-reviewers champion it.

**Paper Topic And Main Contributions:**

The paper is about improving Chinese pop song and Hokkien Gezi Opera singing voice synthesis by enhancing local modeling. The paper proposes two methods to address the problem of local incongruity in Singing Voice Synthesis (SVS) based on Transformer. Local incongruity is characterized by local pronunciation jitter or local noise in the synthesized audio. The first method is to devise a nearest neighbor local attention that only focuses on the adjacent phoneme tokens located before and after the current phoneme token. The local attention representation is fused with the global self-attention representation using a gated unit mechanism. The second method is to design a phoneme-level local adaptive weights loss that optimizes the hard-to-synthesize parts of the predicted mel-spectrogram. The loss function calculates the phoneme-level adaptive confidence scores based on the distance between the predicted and real mel-spectrogram, and normalizes them to adaptive weights.

**Questions For The Authors:**

1. The paper relies on a pre-trained HiFi-GAN vocoder to synthesize audio from the predicted mel-spectrogram. However, HiFi-GAN is trained on natural speech data, which may not be suitable for singing voice synthesis. It would be interesting to see how the proposed methods perform with a vocoder that is specifically trained on singing voice data.
2. How do the proposed methods handle the cases where the phoneme duration is very long or very short, or where the phoneme boundaries are not clear?

**Reasons To Accept:**

The paper conducts extensive experiments on publicly available Chinese pop song and Hokkien Gezi Opera datasets. The results show that the proposed methods achieve significant improvements in both objective and subjective evaluations compared to the strong baseline. The methods can also alleviate the local incongruity problem and improve the quality of synthesized audio.

**Reasons To Reject:**

The paper only focuses on Chinese pop song and Hokkien Gezi Opera datasets, which are relatively small and specific. It is not clear how well the proposed methods can generalize to other languages, genres, and styles of singing voice synthesis.
The paper does not provide a detailed analysis of the local incongruity problem, such as the causes, types, and frequency of local pronunciation jitter or noise. It also does not compare the proposed methods with other existing methods that aim to address the same problem, such as PostNet (Lee et al., 2021).

**Reproducibility:**

3: Could reproduce the results with some difficulty. The settings of parameters are underspecified or subjectively determined; the training/evaluation data are not widely available.

**Reviewer Confidence:**

4: Quite sure. I tried to check the important points carefully. It's unlikely, though conceivable, that I missed something that should affect my ratings.

---

> ### Author Rebuttal · Authors · 2023-08-28
>
> **Q1**:The paper relies on a pre-trained HiFi-GAN vocoder to synthesize audio from the predicted mel-spectrogram. However, HiFi-GAN is trained on natural speech data, which may not be suitable for singing voice synthesis. It would be interesting to see how the proposed methods perform with a vocoder that is specifically trained on singing voice data.
>
> **A1**:Thank you for your question. Sorry for the inaccurate expression of the vocoder used in our paper. The HiFi-GAN vocoder we used is actually a HiFi-GAN singing model pre-trained by the DiffSinger open-source project, which is specially designed for singing voice synthesis. They trained HiFi-GAN singing model using singing data instead of speech data.
>
> **Q2**:How do the proposed methods handle the cases where the phoneme duration is very long or very short, or where the phoneme boundaries are not clear?
>
> **A2**:Thank you for your question.
> Firstly, our method still performs well even when the phoneme duration is very long or very short. In the Gezi Opera dataset, the longest phoneme duration is nearly 10 seconds, and the shortest is a few tenths of a second. After the experiment, we listened to the synthesized samples and found that even in these situations, the synthesized samples performance was good. So our method is not affected by the phoneme duration being too long or too short. Our method has good robustness.
> Secondly, the situation where the phoneme boundary you mentioned is not clear does not exist in our research. Because in the data processing stage, we first need to align the phonemes with the audio in terms of time. Once aligned, we assume that the phoneme boundaries are clear. The phoneme boundary you mentioned is not clear, which means that the alignment model cannot clearly segment the phoneme duration, which was not covered in this research.
>
> **Q3**:The paper only focuses on Chinese pop song and Hokkien Gezi Opera datasets, which are relatively small and specific. It is not clear how well the proposed methods can generalize to other languages, genres, and styles of singing voice synthesis.
>
> **A3**:Thank you for your question. Firstly, our current paper does only demonstrate significant effects on Chinese pop song and Hokkien Gezi Opera datasets. In fact, we have also verified the performance on the Beijing Opera dataset, but due to some copyright and privacy issues of Beijing Opera dataset, we can not display it. After we solve this problem, we will also add the results soon. The above can only indicate that it is effective in Chinese and its dialects, as well as in popular songs and opera styles. Secondly, to our knowledge, there are currently public datasets for singing voice synthesis in English, Japanese, and Korean, and we will also verify them based on your suggestions. In addition, we also conducted experiments on the English speech synthesis dataset LJ Speech, and the subjective and objective experimental results showed that our method is effective.
>
> **Q4**:The paper does not provide a detailed analysis of the local incongruity problem, such as the causes, types, and frequency of local pronunciation jitter or noise.
>
> **A4**:Thank you for your question. As you mentioned, N-Singer (Lee et al., 2021) is indeed a model for solving similar problems. They mentioned in their paper that 'there are word skipping and repetition problems in the synthesized singing voices because of incorrect attention alignment'. From their conclusion, it can be seen that word skipping and repetition problems are due to incorrect attention alignment. Strictly speaking, they are solving word skipping and repetition problems, while we are solving local pronunciation jitter and noise problems. Actually, these are two different problems. We were inspired by them and analyzed our problem from the perspective of attention mechanism. We visualized and analyzed the global self-attention matrix of the first Transformer block in the FFT-Singer decoder, as shown in Figure 2. We found deficiencies in the global self-attention mechanism, so we made improvements. When we wrote the paper, the problem analysis section did not mention why we analyzed the problem from the perspective of attention mechanism. We will supplement the conclusion in the N-Singer.
>
> **Q5**:It also does not compare the proposed methods with other existing methods that aim to address the same problem, such as PostNet (Lee et al., 2021).
>
> **A5**:Thank you for your question. We did not compare our results with the results of N-Singer (Lee et al., 2021) in our paper. The innovation of N-Singer lies in the addition of PostNet to the predicted spectrum and the use of GAN training. N-Singer conducted experiments on Korean datasets. In fact, in our previous research, we found that N-Singer did not have open source code, and we actually reproduced it according to the description in the paper. However, we are not sure if our reproduction is completely consistent with the original paper, so we did not easily release the results. Below is an objective and subjective comparison between our results and N-Singer's results in PopCS and Gezi Opera dataset.
> #### **Objective results in PopCS dataset.**
>
> | Model | MCD(dB)$\downarrow$ | MSD(dB)$\downarrow$ | GPE(\%)$\downarrow$ | VDE(\%)$\downarrow$ | FFE(\%)$\downarrow$ |
> | --- | --- | --- | --- | --- | --- |
> | Baseline | 3.4065 | 1.7164 | 0.74 | 3.63 | 4.05 |
> | Baseline-T+C| 3.2646 | 1.638 | 0.75 | 3.75 | 4.18 |
> | Baseline-T+C+A | 3.062 | 1.5475 | 0.75 | 3.46 | 3.89 |
> | Baseline-T+C+L | 2.9991 | 1.5452 | **0.64** | 3.83 | 4.2 |
> | Baseline-T+C+A+L | **2.8735** | **1.4809** | 0.65 | **3.3** | **3.67** |
> | N-Singer | 2.9561 | 1.5523 | 3.29 | 3.85 | 4.95 |
>
> #### **Subjective results in PopCS dataset.**
>
> | Model | MOS$\uparrow$ |
> | --- | --- |
> | Ground Truth | 4.43$\pm$0.08 |
> | Baseline | 3.55$\pm$0.12 |
> | Baseline-T+C+A+L | **3.71**$\pm$0.11 |
> | N-Singer | 3.65$\pm$0.1 |
>
> #### **Objective results in Gezi Opera dataset.**
>
> | Model | MCD(dB)$\downarrow$ | MSD(dB)$\downarrow$ | GPE(\%)$\downarrow$ | VDE(\%)$\downarrow$ | FFE(\%)$\downarrow$ |
> | --- | --- | --- | --- | --- | --- |
> | Baseline | 3.4694 | 1.7498 | 1.5 | 3.7 | 4.81 |
> | Baseline-T+C| 3.3911 | 1.6924 | 1.75 | 3.67 | 4.97 |
> | Baseline-T+C+A | 3.0314 | 1.5459 | 1.75 | 3.78 | 5.07 |
> | Baseline-T+C+L | 3.017 | 1.5595 | 1.47 | 3.59 | 4.71 |
> | Baseline-T+C+A+L | **2.931** | **1.5144** | **1.34** | **3.57** | **4.57** |
> | N-Singer | 3.021 | 1.582 | 5.88 | 3.69 | 7.82 |
>
> #### **Subjective results in Gezi Opera dataset.**
>
> | Model | MOS$\uparrow$ |
> | --- | --- |
> | Ground Truth | 4.33$\pm$0.09 |
> | Baseline | 3.46$\pm$0.15 |
> | Baseline-T+C+A+L | **3.61**$\pm$0.12 |
> | N-Singer | 3.58$\pm$0.11 |
>
> From the above four tables, it can be seen that our model outperforms N-Singer model in both objective and subjective evaluation metrics on the PopCS and Gezi Opera dataset.

---

### Official Review · Reviewer_cHLv · 2023-08-10

**Soundness:** 3

**Excitement:**

4: Strong: This paper deepens the understanding of some phenomenon or lowers the barriers to an existing research direction.

**Paper Topic And Main Contributions:**

This paper proposes two methods to enhance local modeling in the acoustic model of singing voice synthesis, including a nearest neighbor local attention mechanism and a proposed phoneme-level local adaptive weights loss function.  In the nearest neighbor local attention mechanism, each phoneme token focuses only on the adjacent phoneme tokens located before and after it. Additionally, the phoneme-level local adaptive weights loss function enables the model to focus more on the hard-to-synthesize parts of the mel-spectrogram. The evaluations on the publicly available Chinese pop song and Hokkien Gezi Opera datasets showed the effectiveness and universality of the proposed model.

**Questions For The Authors:**

in Section 4.3, it is not clear how to make the proposed method effective on the Diffsinger, which is a diffusion model.

**Reasons To Accept:**

The paper proposes a nearest neighbor local attention mechanism and a proposed phoneme-level local adaptive weights loss function to enhance local modeling. The experiments have demonstrated the simplicity and effectiveness

**Reasons To Reject:**

The paper claims to address the problem of local modeling in the transformer-based acoustic model. Conformer[1] has been proven to be very effective in modeling both local and global dependencies by combining convolution neural networks and transformers. However, the paper doesn't compare the proposed method with Conformer.

[1] Gulati A, Qin J, Chiu C C, et al. Conformer: Convolution-augmented Transformer for Speech Recognition[J]. Interspeech 2020, 2020.

**Reproducibility:**

3: Could reproduce the results with some difficulty. The settings of parameters are underspecified or subjectively determined; the training/evaluation data are not widely available.

**Reviewer Confidence:**

3: Pretty sure, but there's a chance I missed something. Although I have a good feel for this area in general, I did not carefully check the paper's details, e.g., the math, experimental design, or novelty.

---

> ### Author Rebuttal · Authors · 2023-08-27
>
> **Q1**:in Section 4.3, it is not clear how to make the proposed method effective on the Diffsinger, which is a diffusion model.
>
> **A1**:DiffSinger is an acoustic model for singing voice synthesis based on the diffusion probabilistic model. When training and inference, the auxiliary encoder-decoder model FFT-Singer is an input for DiffSinger. During training, the output of the FFT-Singer encoder is an input of the DiffSinger denoiser. During inference, the predicted mel-spectrogram by the FFT-Singer decoder is an input to DiffSinger. The DiffSinger model adds noise to predicted mel-spectrogram and then trains the denoiser model to predict the noise. Our local modeling enhancement method effectively improves the performance of FFT-Singer, so we replaced the auxiliary encoder-decoder model of DiffSinger with our model instead of FFT-Singer. These two situations are the two diffusion conditions mentioned in the paper. One diffusion condition is the mel-spectrogram predicted by FFT-Singer and the other diffusion condition is the mel-spectrogram predicted by FFT-Singer added our methods. We have all retrained DiffSinger's denoiser based on these two conditions. The experimental results are shown in Table 4, the audio synthesized by DiffSinger on the basis of FFT-Singer added our methods is better.
>
> **Q2**:The paper claims to address the problem of local modeling in the transformer-based acoustic model. Conformer[1] has been proven to be very effective in modeling both local and global dependencies by combining convolution neural networks and transformers. However, the paper doesn't compare the proposed method with Conformer.
>
> **A2**:In fact, we compared our proposed method with Conformer (Baseline-T+C). As shown in Table 1, Baseline-T+C is the replacement of the baseline model decoder from Transformer to Conformer. Baseline-T+C+A, Baseline-T+C+L and Baseline-T+C+A+L are three scenarios where our proposed methods are added to the Conformer. We can see that Conformer is effective as a decoder, and with the addition of the two methods we proposed, the effect has been further improved.

---

### Official Review · Reviewer_SqYD · 2023-08-10

**Soundness:** 3

**Excitement:**

3: Ambivalent: It has merits (e.g., it reports state-of-the-art results, the idea is nice), but there are key weaknesses (e.g., it describes incremental work), and it can significantly benefit from another round of revision. However, I won't object to accepting it if my co-reviewers champion it.

**Paper Topic And Main Contributions:**

This paper aims to improve the local modeling ability of the acoustic model in the field of singing voice synthesis. It propose two local modeling enhancement methods, a nearest neighbor local attention and a phoneme-level local adaptive weights loss function which effectively enhance local modeling for mel-spectrogram prediction and greatly alleviate the local incongruity problem in SVS tasks.

**Questions For The Authors:**

In Section 4.4, “Figure 5: The mel-spectrogram visualization of same sample in the Gezi Opera dataset”, the mel-spectrogram predictions in Figure 5b and Figure 5c look fairly similar, with no noticeable improvement in Figure 5c generated by the proposed method compared to Figure 5b from the baseline model. So it seems the proposed method may not significantly enhance the predictions beyond the baseline in this case, unless I am missing some finer details.

**Reasons To Accept:**

1. Demonstrate a comprehensive understanding of earlier studies in SVS tasks through careful summarization.
2. Present two easy approaches to tackle the challenges of local modeling and local-based enhancement for singing voice synthesis. Fuse local and global representation, introduce a phoneme-level local adaptive weights loss function focusing more on the hard-to-synthesize parts of the mel-spectrogram.
3. The Method section provides a thorough and clear explanation of each part of the model in detail.
4. A multitude of ablation experiments were conducted to underscore the effectiveness of the proposed methods.

**Reasons To Reject:**

1. The introduction briefly discusses some prior methods for enhancing local modeling in singing voice synthesis, such as N-singer and adversarial training methods. It is necessary to include a comparative analysis of these methods with the proposed approach in the experiments. Merely relying on subjective MOS experiments comparing with DiffSinger is insufficient; additional objective experimental comparisons should be incorporated.
2. It is customary to provide a demonstration webpage for SVS work, as subjective listening is necessary to demonstrate its superiority. Although the author has included a few samples in the supplementary material, the limited quantity makes it difficult to convince readers.
3. This paper is presented with clarity and simplicity, but it lacks innovation. The introduction of a mask matrix to enhance local attention and modifications in the loss function are common tricks that are not novel. These modifications could potentially be implemented with just a few lines of code. Moreover, it is crucial to note that the methods proposed in the paper do not fully address the issue of local incongruity in SVS, which limits the potential impact of this work.

**Reproducibility:**

4: Could mostly reproduce the results, but there may be some variation because of sample variance or minor variations in their interpretation of the protocol or method.

**Reviewer Confidence:**

4: Quite sure. I tried to check the important points carefully. It's unlikely, though conceivable, that I missed something that should affect my ratings.

---

> ### Author Rebuttal · Authors · 2023-08-28
>
> **Q1**:In Section 4.4, “Figure 5: The mel-spectrogram visualization of same sample in the Gezi Opera dataset”, the mel-spectrogram predictions in Figure 5b and Figure 5c look fairly similar, with no noticeable improvement in Figure 5c generated by the proposed method compared to Figure 5b from the baseline model. So it seems the proposed method may not significantly enhance the predictions beyond the baseline in this case, unless I am missing some finer details.
>
> **A1**:Thank you for your question. Figure 5c and Figure 5b do not show significant improvement in the red box section. The mel-spectrogram in the example in this paper was not exported as a vectorgraph, and the insertion into the paper resulted in poor performance. We have corrected this issue. Through Figure 5, we want to illustrate that even in non breathing or silent segments, our model has some improvement compared to the baseline model. We can see that Figure 5c shows a significant improvement in the breathing or silent (voiceless) segments compared to Figure 5b. In the voiced segments, there is also an improvement in the medium and high frequency section, where the mel-spectrogram is more detailed and not as blurry as Figure 5b. In summary, we found that our model has significant improvements in the breathing or silent (voiceless) segment, but not particularly significant in the voiced segment. This is because the original model actually has a relatively serious problem of local incongruity in the breathing or silent (voiceless) segments, while the voiced segments are not serious.
>
> **Q2**:The introduction briefly discusses some prior methods for enhancing local modeling in singing voice synthesis, such as N-singer and adversarial training methods. It is necessary to include a comparative analysis of these methods with the proposed approach in the experiments. Merely relying on subjective MOS experiments comparing with DiffSinger is insufficient; additional objective experimental comparisons should be incorporated.
>
> **A2**:Thank you for your question.
> Firstly, we have added a comparative experiment of N-singer and adversarial training methods. The innovation of N-Singer lies in the addition of PostNet to the predicted spectrum and the use of GAN training. N-Singer conducted experiments on Korean datasets. In fact, in our previous research, we found that N-Singer did not have open source code, and we actually reproduced it according to the description in the paper. However, we are not sure if our reproduction is completely consistent with the original paper, so we did not easily release the results. Below is an objective and subjective comparison between our results, N-Singer model results and adversarial training methods (Baseline+GAN) results in PopCS and Gezi Opera dataset.
>
> | Model | MCD(dB)$\downarrow$ | MSD(dB)$\downarrow$ | GPE(\%)$\downarrow$ | VDE(\%)$\downarrow$ | FFE(\%)$\downarrow$ |
> | --- | --- | --- | --- | --- | --- |
> | Baseline | 3.4065 | 1.7164 | 0.74 | 3.63 | 4.05 |
> | Baseline-T+C| 3.2646 | 1.638 | 0.75 | 3.75 | 4.18 |
> | Baseline-T+C+A | 3.062 | 1.5475 | 0.75 | 3.46 | 3.89 |
> | Baseline-T+C+L | 2.9991 | 1.5452 | **0.64** | 3.83 | 4.2 |
> | Baseline-T+C+A+L | **2.8735** | **1.4809** | 0.65 | **3.3** | **3.67** |
> | N-Singer | 2.9561 | 1.5523 | 3.29 | 3.85 | 4.95 |
> | Baseline+GAN | 3.0324 | 1.645 | 5.88 | 4.05 | 8.81 |
> #### **Subjective results in PopCS dataset.**
>
> | Model | MOS$\uparrow$ |
> | --- | --- |
> | Ground Truth | 4.43$\pm$0.08 |
> | Baseline | 3.55$\pm$0.12 |
> | Baseline-T+C+A+L | **3.71**$\pm$0.11 |
> | N-Singer | 3.65$\pm$0.1 |
> | Baseline+GAN | 3.63$\pm$0.13 |
>
> #### **Objective results in Gezi Opera dataset.**
>
> | Model | MCD(dB)$\downarrow$ | MSD(dB)$\downarrow$ | GPE(\%)$\downarrow$ | VDE(\%)$\downarrow$ | FFE(\%)$\downarrow$ |
> | --- | --- | --- | --- | --- | --- |
> | Baseline | 3.4694 | 1.7498 | 1.5 | 3.7 | 4.81 |
> | Baseline-T+C| 3.3911 | 1.6924 | 1.75 | 3.67 | 4.97 |
> | Baseline-T+C+A | 3.0314 | 1.5459 | 1.75 | 3.78 | 5.07 |
> | Baseline-T+C+L | 3.017 | 1.5595 | 1.47 | 3.59 | 4.71 |
> | Baseline-T+C+A+L | **2.931** | **1.5144** | **1.34** | **3.57** | **4.57** |
> | N-Singer | 3.021 | 1.582 | 5.88 | 3.69 | 7.82 |
> | Baseline+GAN | 3.0728 | 1.937 | 6.94 | 4.12 | 9.86 |
> #### **Subjective results in Gezi Opera dataset.**
>
> | Model | MOS$\uparrow$ |
> | --- | --- |
> | Ground Truth | 4.33$\pm$0.09 |
> | Baseline | 3.46$\pm$0.15 |
> | Baseline-T+C+A+L | **3.61**$\pm$0.12 |
> | N-Singer | 3.58$\pm$0.11 |
> | Baseline+GAN | 3.51$\pm$0.12 |
>
> From the above four tables, it can be seen that our model outperforms N-Singer or Baseline+GAN in both objective and subjective evaluation metrics on the PopCS and Gezi Opera dataset. We found that the GAN-based method performed poorly on GPE, VDE and FFE these three metrics. N-Singer performs better than Baseline+GAN, because PostNet in N-Singer is a convolutional network that can pay some attention to local features.
>
> Secondly, we have added objective evaluation results for Section4.3 as shown in the table below .
> #### **Objective evaluation results.**
>
> | Dataset | Model | MCD(dB)$\downarrow$ | MSD(dB)$\downarrow$ | GPE(\%)$\downarrow$ | VDE(\%)$\downarrow$ | FFE(\%)$\downarrow$ |
> | --- | --- | --- | --- | --- | --- | --- |
> | PopCS | DiffSinger | 3.1452 | 1.7365 | **0.72** | **3.67** | **4.08** |
> | PopCS | DiffSinger+Our | **3.0564** | **1.6706** | 0.9 | 3.74 | 4.24 |
> | Gezi Opera | DiffSinger | 3.9807 | 2.0012 | 1.7 | 3.84 | 5.1 |
> | Gezi Opera | DiffSinger+Our | **3.9145** | **1.9815** | **1.64** | **3.67** | **4.89** |
>
> From the table, we can see that adding our method to DiffSinger resulted in a decrease in both MCD and MSD in the PopCS dataset, while there was almost no significant change in GPE, VDE and FFE. In the Gezi Opera dataset, all metrics have shown a certain decrease. Combining subjective and objective evaluation metrics, the overall improvement effect for DiffSinger is not significant. The actual situation is indeed so, because the DiffSinger model itself has good performance, and our method was not originally designed to improve the performance of the DiffSinger model. Although this method has a small improvement on the DiffSinger model, we did significantly improve the performance of FFT-Singer, which is the starting point of this study. Our purpose in conducting this experiment is to verify that our method is flexible and effective even when combined with the DiffSinger model.
>
> **Q3**:It is customary to provide a demonstration webpage for SVS work, as subjective listening is necessary to demonstrate its superiority. Although the author has included a few samples in the supplementary material, the limited quantity makes it difficult to convince readers.
>
> **A3**:Thank you for your question. A demonstration webpage is indeed necessary. We have created a demonstration webpage, but due to the fact that rebuttal does not allow adding links, we are unable to present it to you at the moment.
>
> **Q4**:This paper is presented with clarity and simplicity, but it lacks innovation. The introduction of a mask matrix to enhance local attention and modifications in the loss function are common tricks that are not novel. These modifications could potentially be implemented with just a few lines of code. Moreover, it is crucial to note that the methods proposed in the paper do not fully address the issue of local incongruity in SVS, which limits the potential impact of this work.
>
> **A4**:Your evaluation is very relevant. The improvement workload for the FFT-Singer model in this paper is a bit small, but the two improvement methods proposed in this paper are both simple and effective, and our methods have good robustness and generalization. As far as we know, we constructed the Gezi Opera dataset for the first time in singing voice synthesis research, and the process of constructing the dataset was very difficult and time-consuming. In addition, we validated the effectiveness of the methods and the effectiveness of the Gezi Opera dataset. We have truly expanded the scope of research on singing voice synthesis. To be honest, the method proposed in this paper has not yet fully solved the problem of local incongruity, but this is only the beginning of our research and we are still exploring.

---

### Meta-Review · Area_Chair_FcFE · 2023-09-06

**Recommendation:** 3

**Metareview:**

**Summary:**
The paper presents two methods to improve local modelling to address the issue of local incongruity in the context of singing voice synthesis (SVS):
1)	A nearest neighbour local attention mechanism to enhancing local modelling by incorporating adjacent phoneme tokens' information.
2)	A phoneme-level local adaptive weights loss function, targeting the harder-to-synthesize parts of Mel-spectrograms.
The paper conducts experiments on Chinese pop songs and Hokkien Gezi Opera datasets to validate the effectiveness of the proposed methods.


**Pros:**

- The paper effectively summarizes prior work on SVS task.

- The proposed methods are described clearly, and their effects are demonstrated by ablation experiments.

- The proposed methods consistently outperform alternatives methods (N-Singer and adversarial methods) in both objective and subjective evaluation metrics.

- The authors provide a detailed rebuttal addressing reviewers' questions and concerns, effectively clarifying their approach and results.


**Cons:**
- Reviewers 2 and 4 found the innovations presented in the paper to be relatively small and simplistic, mostly involving integrating local attention with global attention and adapting loss weights.

- While the paper addresses concerns about adding a comparative analysis with N-Singer and adversarial training methods, it lacks a detailed analysis of local incongruity and comparison with existing methods (Reviewer 3, Reviewer 4).

- The methods' effectiveness is primarily demonstrated on specific datasets (Chinese pop songs and Hokkien Gezi Opera), raising questions about the generalization of these methods to other languages, genres, and styles of singing (Reviewer 3).

- The suggested dataset lacks both open-source availability and comprehensive explanations, which hinders its ability to facilitate progress in the field (Reviewer 1).



Reviewer 1: SqYD
Reviewer 2: cHLv
Reviewer 3: DiTD
Reviewer 4: knLR

---

### Decision · Program_Chairs · 2023-10-07

**Decision:**

Accept-Main

**Comment:**

**Summary:**
The paper presents two methods to improve local modelling to address the issue of local incongruity in the context of singing voice synthesis (SVS):
1)	A nearest neighbour local attention mechanism to enhancing local modelling by incorporating adjacent phoneme tokens' information.
2)	A phoneme-level local adaptive weights loss function, targeting the harder-to-synthesize parts of Mel-spectrograms.
The paper conducts experiments on Chinese pop songs and Hokkien Gezi Opera datasets to validate the effectiveness of the proposed methods.


**Pros:**

- The paper effectively summarizes prior work on SVS task.

- The proposed methods are described clearly, and their effects are demonstrated by ablation experiments.

- The proposed methods consistently outperform alternatives methods (N-Singer and adversarial methods) in both objective and subjective evaluation metrics.

- The authors provide a detailed rebuttal addressing reviewers' questions and concerns, effectively clarifying their approach and results.


**Cons:**
- Reviewers 2 and 4 found the innovations presented in the paper to be relatively small and simplistic, mostly involving integrating local attention with global attention and adapting loss weights.

- While the paper addresses concerns about adding a comparative analysis with N-Singer and adversarial training methods, it lacks a detailed analysis of local incongruity and comparison with existing methods (Reviewer 3, Reviewer 4).

- The methods' effectiveness is primarily demonstrated on specific datasets (Chinese pop songs and Hokkien Gezi Opera), raising questions about the generalization of these methods to other languages, genres, and styles of singing (Reviewer 3).

- The suggested dataset lacks both open-source availability and comprehensive explanations, which hinders its ability to facilitate progress in the field (Reviewer 1).



Reviewer 1: SqYD
Reviewer 2: cHLv
Reviewer 3: DiTD
Reviewer 4: knLR